

# Insights into the O:C dependent mechanisms controlling the evaporation of α-pinene secondary organic aerosol particles

Angela Buchholz[1], Andrew T. Lambe[2], Arttu Ylisirniö[1], Zijun Li[1], Olli-Pekka Tikkanen[1], Celia Faiola[3], Eetu Kari[1], Liqing Hao[1], Olli Luoma[1], Wei Huang[4], Claudia Mohr[5], Douglas R. Worsnop[1,2], Sergey A. Nizkorodov[6], Taina Yli-Juuti[1], Siegfried Schobesberger[1], Annele Virtanen[1]

[1]Department of Applied Physics, University of Eastern Finland, Kuopio, Finland
[2]Aerodyne Research Inc., Billerica, MA 08121-3976, USA
[3]Department of Ecology and Evolutionary Biology, University of California, Irvine, CA, USA
[4]Institute of Meteorology and Climate Research, Karlsruhe Institute of Technology, Karlsruhe, Germany
[5]Department of Environmental Science and Analytical Chemistry, Stockholm University, Stockholm, Sweden
[6]Department of Chemistry, University of California, Irvine, CA, USA

## Abstract

The volatility of oxidation products of volatile organic compounds (VOCs) in the atmosphere is a key factor to determine if they partition into the particle phase contributing to secondary organic aerosol (SOA) mass. Thus, linking volatility and measured particle composition will provide insights into SOA formation and its fate in the atmosphere. We produced α-pinene SOA with three different oxidation levels (characterised by average oxygen to carbon ratio, $O:C$ = 0.53, 0.69, and 0.96) in an oxidation flow reactor. We investigated the particle volatility by isothermal evaporation in clean air as a function of relative humidity (RH < 2%, 40%, and 80%) and used a filter-based thermal desorption method to gain volatility and chemical composition information.

We observed reduced particle evaporation for particles with increasing $O:C$ ratio, indicating that particles become more resilient to evaporation with oxidative aging. Particle evaporation was increased in the presence of water vapour and presumably particulate water, at the same time the resistance of the residual particles to thermal desorption was increased as well. For SOA with $O:C$ = 0.96, the unexpectedly large increase of mean thermal desorption temperature and changes in the thermogram shapes under wet conditions (RH80%) were an indication of aqueous phase chemistry. For the lower $O:C$ cases, some water induced composition changes were observed. However, the enhanced evaporation under wet conditions could be explained by the reduction of particle viscosity from the semi-solid to liquid-like range and the observed higher desorption temperature of the residual particles is a direct consequence of the increased removal of high volatility and remaining of low volatility compounds.

## 1 Introduction

Secondary organic aerosol (SOA) accounts for a major fraction of the global atmospheric aerosol burden (Hallquist et al., 2009; Jimenez et al., 2009). Understanding the mechanism of formation and properties of SOA is therefore of utmost



importance to estimate its effects on climate, air quality and human health. SOA – by definition – is formed when low volatility oxidation products of volatile organic compounds (VOCs) deposit onto existing particles or form new particles. While the scientific community has made significant improvements in characterising the precursors and gas phase oxidation products at the molecular level with new measurement techniques (e.g. Ehn et al., 2014; Lopez-Hilfiker et al., 2014; Mohr et al., 2017),

the particle phase is still proving to be complicated. There are technical challenges as chemical composition analysis techniques typically operate with liquid or gaseous samples, and thus aerosol particles samples need to be extracted or desorbed which can introduce artefacts. In addition, there is a multitude of compounds in aerosol particles (Goldstein and Galbally, 2007), and the particle phase is far from static once it is formed. Many different chemical processes can occur in the particle phase, especially if water is present, e.g., larger molecules may be formed by polymerisation reactions or there may be oxidation with

oxidants taken up from the gas phase (Herrmann et al., 2015; Kroll and Seinfeld, 2008). In addition, aerosol particles interact with the gas phase constantly. Compounds will partition between gas and particle phase depending e.g. on their vapour pressures and concentrations in the different phases (Donahue et al., 2006).

Gas-particle partitioning has been described in detail and applied to model SOA formation (Pankow, 1994a, 1994b; Pankow et al., 2001). Donahue et al. (2006) bypassed the issue of having to define the phase diagram for each compound in the SOA

system by introducing the volatility basis set (VBS) framework which utilises the effective saturation concentration (C*) to group the compounds into volatility classes (or bins). This framework can then be used to follow changes in partitioning, and thus bulk volatility, with time and to predict SOA formation, e.g. in global modelling (e.g. Bergström et al., 2012; Jimenez et al., 2009). More comprehensive schemes have since been developed linking volatility with other properties, e.g., oxidation level represented by elemental oxygen-to-carbon ($\overline{O:C}$) ratio (Donahue et al., 2011, 2012) or a certain functionality (Krapf et

al., 2016).

In principle, VBS distributions can be derived from any type of volatility measurements. Besides the SOA yield studies, the most common approaches are to measure the evaporation of SOA particles at elevated temperatures (typically 30 – 300 °C) or by removing the gas phase around the particles through isothermal dilution. Thermal methods include measurements of particle mass loss after passing through a thermodenuder (An et al., 2007; Cappa and Wilson, 2011; Huffman et al., 2008; Kolesar et

al., 2015) and methods where particles are collected and then desorbed with a heated carrier gas flow (e.g. the filter inlet for gases and aerosols sampling unit, FIGAERO, Lopez-Hilfiker et al., 2014). Typically, the remaining SOA mass or volume fraction is determined, or the changes in chemical composition with temperature are monitored. With any thermal method, there is the potential of thermal decomposition at elevated temperatures prior to detection and some studies report strong indications of decomposition and particle phase reactions (Hall IV and Johnston, 2012; Kolesar et al., 2015; Lopez-Hilfiker et

al., 2015; Stark et al., 2017). This may lead to an overestimation of the (semi-)volatile fraction of the SOA (Lopez-Hilfiker et al., 2015; Schobesberger et al., 2018). This issue may be constrained by combining isothermal dilution and thermodenuder measurements (Grieshop et al., 2009; Louvaris et al., 2017).

Vaden et al. (2011) proposed an isothermal dilution method utilising a differential mobility analyser (DMA) for particle size selection coupled to an evaporation/residence time chamber at room temperature with active carbon as an absorber for the



evaporating vapours. This method avoids issues associated with thermal decomposition in thermodenuders and wall loss artefacts in smog chamber experiments. It enables studying evaporation times of up to two days. With this method, it was shown that dry particles evaporate much slower than expected from SOA yield experiments (Vaden et al., 2011). Combining this type of data with detailed process modelling suggested that mass transport limitations in the most likely semi-solid, and thus highly viscous, dry particles were responsible for the observed slow evaporation (Yli-Juuti et al., 2017). In this context, particulate water can play a crucial role as it can act as a plasticiser, reducing particle viscosity and thus enhance evaporation (Wilson et al., 2015; Yli-Juuti et al., 2017). Aqueous phase chemical reactions may also have an impact on SOA volatility, enhancing formation of larger molecules or, conversely, hydrolysing especially peroxy compounds. To distinguish between the physical (viscosity decrease) and chemical (aqueous phase reactions) role of particulate water in particle evaporation, it is necessary to investigate the chemical composition of SOA particles during isothermal evaporation at ambient temperatures under a range of RH conditions. Although the thermal desorption in FIGAERO-CIMS may cause decomposition as described above, it still can be useful for this purpose by combining measurements of fresh SOA particles and of particles that have been left to evaporate for long times. Whereas D'Ambro et al. (2018) left the collected SOA particles on the FIGAERO filter in a clean nitrogen flow for up to 24 h, we utilised our residence time chamber (RTC) to probe the particle phase composition after different evaporation times.

In this study we investigated the volatility of SOA particles formed from α-pinene at different oxidation levels with a combination of isothermal evaporation and thermal desorption. In addition, we monitored the evolution of the chemical composition of the residual particle after evaporation using FIGAERO-CIMS. Special focus was on the role of particulate water on the evaporation behaviour to determine whether water simply affected the rate transfer through the particles or also induced chemical reactions in the particles. Understanding particle phase processes is crucial to improving our ability to model SOA formation and to predict its behaviour in the atmosphere.

## 2    Methods

### 2.1    Measurements

The experiments were conducted in the same fashion as described in Yli-Juuti et al., 2017 with the main difference that here SOA was formed in a potential aerosol mass reactor (PAM, Aerodyne Research Inc., Kang et al., 2007; Lambe et al., 2011) through ozonolysis and photooxidation of α-pinene instead of pure ozonolysis. A schematic of the experimental setup is shown in Figure S1 in the Supplement Information. α-Pinene vapour was introduced to the PAM reactor by flowing nitrogen over a vial of α-pinene (≥ 98% purity, Sigma Aldrich) placed in a diffusion source resulting in a mixing ratio of 190 parts per billion by volume (ppb). The α-pinene mixing ratio input to the PAM reactor was 190 ppb as monitored with a high-resolution time-of-flight proton transfer reaction mass spectrometer (PTR-MS, Ionicon model 8000). Concentrations of α-pinene were checked periodically at the outlet of the PAM reactor and were < 1 ppb for all settings (i.e., the precursor reacted completely). The PAM reactor was operated at a constant temperature of 27 ℃ and RH of 40%. Hydroxyl radicals (OH) were formed from





ozone ($O_3$) photolysis and reaction of the generated $O(^1D)$ with water vapour inside the PAM reactor. The $O_3$ concentration and the irradiation level in the PAM reactor were varied to create SOA with a range of effective oxidation ages (see Table 1). The polydisperse SOA was characterised with a scanning mobility particle sizer (SMPS, TSI Inc., model 3082+3775) and a high resolution time-of-flight aerosol mass spectrometer (AMS, Aerodyne Research Inc., DeCarlo et al., 2006).

To conduct the isothermal evaporation measurements, two nano differential mobility analysers (NanoDMA, TSI Inc., model 3085) were used in parallel to select a narrow distribution of monodisperse particles (mobility diameter 80 nm). Two NanoDMAs were utilised to double the available sample flow without compromising on the sheath to sample flow ratio. Each NanoDMA was operated with an open loop sheath flow, and the residence time inside the NanoDMA was ≤ 0.3 s, which limits the diffusional mixing of gas phase compounds into the selected sample flow. This led to a high dilution of the gas phase

compounds (by more than one order of magnitude) and resulted in a sudden shift from the gas-particle equilibrium at the NanoDMA outlet, which initiated the particle evaporation. Due to some length of tubing always required for sampling, the shortest particle evaporation time, i.e. the minimum time between exiting the NanoDMA and reaching subsequent measurement devices, was 2 s. Evaporation times of up to 160 s were achieved by simply adding tubing to the sample line and/or reducing the sample flow. Size and composition of the size selected particles were monitored using an SMPS, an AMS,

and a filter inlet for gases and aerosols sampling unit (FIGAERO, Aerodyne Research Inc., Lopez-Hilfiker et al., 2014) in combination with a chemical ionisation mass spectrometer (CIMS, Aerodyne Research Inc., Lee et al., 2014) using iodide as reagent ion. The overall RH during the evaporation experiment was controlled with the sheath flow of the NanoDMAs at three different levels: dry (RH < 2%), RH40%, and RH80%. For RH40% and RH80%, the sheath flow was humidified with a Nafion humidifier (Permapure, model PD-200T).

Longer evaporation times, ranging from 0.5 to 11 h, were achieved by passing the sample flow into a 100 L cylindrically-shaped stainless-steel chamber (Residence Time Chamber, RTC), which had an inlet at the top and an outlet at the bottom. Our earlier study showed that the stainless-steel walls of the evaporation chamber act as nearly perfect sinks for the compounds evaporating from the particles (Yli-Juuti et al., 2017). Before each evaporation experiment, the nanoDMAs, connected tubing, and RTC were flushed for at least 12 h with clean air at the RH of the next experiment. This ensured that all parts of the system

had adjusted to the new conditions. RTC experiments were initiated by adding monodisperse SOA through the top inlet for 20 min while displacing clean air, which exited via the bottom outlet. At the end of filling the RTC, the average particle number concentration was $200 - 1500$ cm$^{-3}$, and the average particle mass concentration was $0.1 - 0.6$ µg/m$^3$. Then the chamber was closed off and periodically opened again to sample for 9 to 15 min intervals with the SMPS and AMS. Clean air with the same RH was admitted into the chamber during sampling intervals to maintain constant pressure and humidity, with corresponding

dilution factors of typically 1.2 or less. As we base our analysis on changes in measured particle size, and not on the change in total particle mass, the particle losses and further dilution only limit the number of times that it is possible to sample from the RTC. The size selection unit, the RTC, and all particle phase measurement instruments were located in a temperature-controlled room (20 °C) to minimise the effect of ambient temperature fluctuations on particle evaporation and on RH. For the



RH80% experiments, the closed loop sheath flow of the SMPS was also humidified to ensure that the RH stayed within +/- 2%

between size selection, evaporation and size measurements.

For FIGAERO-CIMS measurements, the above-described procedure had to be adjusted to accommodate the need for higher

particle mass loadings: (1) "fresh" samples were collected directly after size selection for 20 or 30 min; (2) RTC fill times

5    were increased from 20 to 75 min; and (3) the FIGAERO-CIMS sampled the remaining SOA particles in the RTC once after

to 4 hours of evaporation (data labelled "RTC" in the following). The upper limit of collected mass was estimated from the

particle mass concentration and sampling time. Collected particulate material was 140 - 260 ng for "fresh" and 20 - 70 ng for

RTC samples (Table S2). Due to the collection time needed, the "fresh" filter sample contained particles which had been on

the filter for 0 to 20 or 30 min. As the particle evaporation starts when the gas phase is diluted in the NanoDMAs and no new

particle/gas phase equilibrium can be reached, the evaporation continues while particles are being deposited on the FIGAERO

filter. Thus, some volatile compounds may have already evaporated before the thermal desorption begins and cannot be

captured with this method.

## 2.2    Data analysis

The SMPS data was inverted with the Aerosol Instrument Manager software (TSI). To check the selected sizes of the

NanoDMAs, ammonium sulphate particles, which are non-volatile at room temperature, were sampled for each sheath flow

setting. This actual measured size was then used as "set" size. The Evaporation Factor (EF) was calculated as the ratio of the

measured sizes ($D_{meas}$) and the set sizes ($D_{set}$):

$$EF = \frac{D_{meas}}{D_{set}} \tag{1}$$

Assuming spherical particles, the Volume Fraction Remaining (VFR) can be calculated as:

20                                                     $$VFR = EF^3 \tag{2}$$

In the following, the evolution of VFR as a function of the residence time in the RTC will be called "evapogram".

The high resolution AMS data was analysed with the SQUIRREL (version 1.59D) and PIKA toolkits (version 1.19, DeCarlo

et al., 2006). The improved parameterisation from Canagaratna et al. (2015) was used to perform the elemental analysis which

yields average oxygen-to-carbon and hydrogen-to-carbon ratios for the sampled SOA ($\overline{O:C}$ and $\overline{H:C}$ ratios). The average

carbon oxidation state, $\overline{OSc}$ , was calculated following the approximation in Kroll et al. (2011):

$$\overline{OSc} = 2 \cdot \overline{O:C} - \overline{H:C} \tag{3}$$

Raw data from FIGAERO desorption temperature scans was processed using tofTools, a MATLAB-based software package

developed for analysing ToF-CIMS data (Junninen et al., 2010). The raw data was averaged to provide average mass spectra

spaced by 20 s, and baseline correction was applied before fitting the high-resolution mass spectral data. The average $\overline{O:C}$ and

$\overline{H:C}$ ratios, and composition were calculated as signal weighted sums from the elemental composition defined by the molecular

formulas. Details are given in the SI material.





Thermal desorption of a FIGAERO filter sample via a nitrogen gas flow heated from 25 to 210 °C yields thermograms, i.e., the total or selected ion count rate vs. desorption temperature. The sum over all ions (except the reagent ions) will be referred to as "total thermogram". Where noted, we normalised the thermograms with the time-integral of the respective total thermogram to help compare thermogram shapes. We characterise the thermograms mainly by the temperature of peak

desorption (= thermogram maximum, $T_{max}$), as is common practice (Huang et al., 2018; Lopez-Hilfiker et al., 2014). We also use the median desorption temperature ($T_{median}$), i.e., the temperature dividing the thermogram into two equal areas. This value may reflect the overall desorption characteristics better than $T_{max}$, because thermograms (individual or total) may feature poorly defined peaks and contain large fractions of signal at temperatures very different from (typically higher than) $T_{max}$. Integrated normalised FIGAERO mass spectra were obtained by calculating the time-integral of each ion's count rate over the full

desorption cycle, and then normalising to the sum over all non-reagent ions. The above described normalisation procedures were designed to account for differences between experiments in the amount of particle mass collected on the filter, and it allows us to directly compare the thermogram shapes and the relative contributions of certain ions between different experiments while not affecting $T_{max}$ and $T_{median}$.

In the CIMS instrument, the major class of ions were clusters of iodide ($I^-$) and organic compounds (M) in the sample flow,

detected as $[M+I]^-$. In this study, the voltage settings in the CIMS' ion guidance elements led to a relatively high level of ion declustering, which included the formation of ions not containing iodide and with an odd number of hydrogen atoms likely dominated by $[M+I]^-$ ions that lost HI resulting in $[M-H]^-$ and by other fragmentation processes described in the SI section. These "declustered ions" accounted for 15 – 25% of the total ion signal (see SI for further information). We analysed the data treating the declustered and adduct ions separately. However, for plotting the integrated spectra of all observed species, all

ions were included, and it was assumed that deprotonation to form $[M-H]^-$ was the only declustering reaction. The observed ion formulas were converted into neutral compound formulas by adding the mass of $H^+$ for $[M-H]^-$ or subtracting the mass of $I^-$ for $[M+I]^-$.

## 3    Results and Discussion

### 3.1    SOA chemical composition

For the low, medium and high OH exposure in the PAM reactor, the O:C ratios derived from AMS data of size selected α-pinene SOA were 0.53, 0.69, and 0.96, respectively. These O:C values are representative for fresh and aged ambient SOA in monoterpene rich environments (Aiken et al., 2008; Ng et al., 2010; Ortega et al., 2016). From here on we refer to these three cases as low-, medium-, or high-O:C experiments. Overall, FIGAERO and AMS measurements show the same trends in the O:C ratios, however the AMS derived values show a larger difference in O:C ratios between the low and high OH exposures

(see Table 1). With increasing overall O:C, the FIGAERO mass spectra show an increasing fraction of monomers (defined as compounds with 10 or fewer C atoms, i.e., compounds derivable from a single monoterpene precursor, roughly corresponding to masses < 300 Da) as shown in Figure S2. In the high-O:C case, there is a strong increase in the contribution of smaller

molecules with high O:C (see Figure S2 and Figure S5 e and f), due to the dominance of fragmentation reactions at high OH exposure (Lambe et al., 2012; Palm et al., 2016). It was not expected to find such a large contribution of low molecular weight (MW) compounds such as $C_3H_4O_4$ at 104 Da (detected mostly as $[C_3H_4O_4 + I]^-$) or $C_4H_6O_4$ at 118 Da (detected mostly as $C_4H_5O_4^-$) in the particle phase. Given that the majority of compounds of this size should be too volatile to stay in the particle

phase, a likely cause for the appearance of these low-MW compounds is thermal decomposition of higher MW compounds during the desorption from the FIGAERO filter. Then the increase of low-MW compounds at higher $\overline{OS}_C$ indicates that particulate organics become overall more susceptible to thermal decomposition when SOA is formed under higher OH exposure in the PAM reactor. However, at this point we cannot determine if the increased detection of these low-MW compounds is driven by a higher degree of fragmentation reactions at high OH exposure in PAM or the thermal decomposition

of higher MW compounds in the FIGAERO.

**Table 1: Average oxidation state and average molecular formula derived from FIGAERO and AMS measurements.**

| OH exposure | instrument | sampling conditions | | O:C | H:C | $\overline{OS}_C$ | average composition |
|---|---|---|---|---|---|---|---|
| **low** | **AMS** | | | 0.53 | 1.53 | -0.46 | |
| | **FIGAERO** | **dry** | **fresh** | 0.66 | 1.62 | -0.30 | $C_{9.7} H_{16.0} O_{5.8}$ |
| | | | **RTC** | 0.68 | 1.63 | -0.28 | $C_{10.8} H_{17.9} O_{6.8}$ |
| | | **RH80%** | **fresh** | 0.68 | 1.62 | -0.25 | $C_{10.2} H_{16.8} O_{6.5}$ |
| | | | **RTC** | 0.71 | 1.61 | -0.20 | $C_{10.6} H_{17.4} O_{6.8}$ |
| **medium** | **AMS** | | | 0.69 | 1.42 | -0.05 | |
| | **FIGAERO** | **dry** | **fresh** | 0.75 | 1.52 | -0.03 | $C_{9.0} H_{14.0} O_{6.3}$ |
| | | | **RTC** | 0.74 | 1.53 | -0.04 | $C_{9.3} H_{14.4} O_{6.4}$ |
| | | **RH80%** | **fresh** | 0.76 | 1.52 | 0.00 | $C_{9.0} H_{14.0} O_{6.4}$ |
| | | | **RTC** | 0.77 | 1.55 | -0.02 | $C_{9.8} H_{15.5} O_{6.9}$ |
| **high** | **AMS** | | | 0.96 | 1.26 | 0.63 | |
| | **FIGAERO** | **dry** | **fresh** | 0.84 | 1.46 | 0.23 | $C_{8.2} H_{12.3} O_{6.3}$ |
| | | | **RTC** | 0.83 | 1.47 | 0.19 | $C_{8.4} H_{12.7} O_{6.3}$ |
| | | **RH80%** | **fresh** | 0.85 | 1.43 | 0.27 | $C_{8.0} H_{11.7} O_{6.1}$ |
| | | | **RTC** | 0.84 | 1.46 | 0.22 | $C_{8.5} H_{12.8} O_{6.2}$ |

### 3.2    Linking isothermal evaporation and thermal desorption in FIGAERO

Plots showing the VFR of α-pinene SOA as a function of residence time (i.e., "evapograms") are presented in Fig. 1a - c for the three $\overline{OS}_C$ cases. The dependence of the evaporation rate on RH follows the trends reported in earlier studies: at dry

conditions, the evaporation is substantially slower than at RH40% and RH80% conditions, for all oxidation levels (Vaden et al., 2011; Wilson et al., 2015; Yli-Juuti et al., 2017). In the studies of Wilson et al. (2015) and Yli-Juuti et al. (2017), the slower evaporation under dry conditions was related to increased diffusional limitations, due to higher viscosity than under humid conditions. We note that the evaporation rate at RH40% is higher than at RH80% (Figure 1a - c). This observation is explained by the solution or Raoult effect, i.e., the decrease of the equilibrium vapour pressure over more dilute humidified particles, as

demonstrated by the evaporation modelling presented in the supplemental material (Figure S4) and in the study of Yli-Juuti et



al. (2017). This indicates that diffusion limitations do not play a major role in aerosol evaporation at room temperature when the RH is at atmospherically relevant levels.

The dependence of the isothermal evaporation rate on the oxidation level is reported for the first time in this study. As the O:C ratio of the produced SOA is increasing, the overall rate of evaporation decreases. After 6 h of evaporation, SOA particle volume decreased by only 10% under dry and ~ 40% under RH40% conditions in the high-O:C case. For the low-O:C case, the corresponding numbers are 40% (dry) and 60% (RH40%). These trends suggest that the more highly oxygenated SOA is less volatile, as expected from thermodenuder measurements (e.g., Donahue et al., 2012).

In Figure 1d-f, we show FIGAERO total thermograms (signal-weighted sum of the thermograms for all individual compositions) measured at different time periods during isothermal evaporation at dry and RH80% conditions. The FIGAERO filter sampling periods of each thermogram are marked by coloured boxes in the evapograms in Figure 1a-c. Fresh SOA thermograms were shifted to higher temperatures with increasing O:C, both at dry and RH80% conditions. For the low-, medium-, and high-O:C cases, the peak evaporation temperatures, $T_{max}$, were 50, 60, and 71 °C under dry conditions and 61, 70, and 92 °C at RH80%. These shifts are in line with our isothermal evaporation measurements suggesting a decreasing vapour pressure of SOA compounds with increasing O:C, and are also consistent with a larger role of thermal decomposition during desorption, as indicated by the increased contribution of small highly oxidised molecules discussed above.

When examining the $T_{max}$ of fresh SOA in more detail, it can be seen that at a fixed O:C ratio $T_{max}(RH80\%) > T_{max}(dry)$. This trend can be explained with the evapogram measurements: the particles evaporate more quickly at higher humidity as seen in the evapograms, hence a larger fraction of higher volatility compounds is already lost during the 20 or 30 min period of FIGAERO filter collection before the thermal desorption. Thus, the collected residual particles are less volatile, characterised by a higher $T_{max}$. When sampling from the RTC after longer evaporation time, the VFR is even lower (i.e., a larger volume fraction has been lost due to evaporation). Correspondingly, the thermogram peak is shifted further toward higher temperatures in all studied O:C cases, again indicating an increasing fraction of lower volatility compounds in the residual particles.

Only for the high-O:C case, also an absolute increase in the amount of material desorbing at > 150 °C is observed when comparing the fresh SOA at RH80% and dry conditions (see non-normalised thermograms in Figure S3). Because the estimated organic mass loadings on the filter were comparable, this indicates that when the high-O:C particles, generated in the PAM at RH40%, are exposed to elevated RH (RH80%), compounds with high desorption (and/or decomposition) temperatures are formed in the particle phase. We will corroborate this suggestion below.

In Figure 2, VFR is plotted as a function of $T_{median}$. The figure visualizes two phenomena: Generally, $T_{median}$ and VFR are positively correlated with O:C ratio. As laid out above, this observation is explained by the negative relationship of O:C and volatility. At the same time, however, for a certain O:C ratio, VFR and $T_{median}$ are negatively correlated. As mentioned above, this can be explained by the properties of the residual particles after a certain period of evaporation. We will explore this further now, together with the possibility of water-induced particle phase reactions.

In low- and medium-O:C cases, the trends of the VFR vs $T_{median}$ behaviour are comparable, and the increase in $T_{median}$ is clearly associated with the decreasing VFR, regardless of the RH and hence water content of the particles. The behaviour in the high-



O:C case is different and cannot be explained by the evaporation of higher volatility material alone. For the high-O:C cases, $T_{median}$ of fresh SOA increases from 86 to 104 °C between the dry and RH80% case despite change of only 9% in VFR. Combining this observation with the fact that there is an absolute increase in material desorbed > 150 °C suggests that in the high-O:C cases the particle phase water alters the SOA particle composition, resulting in an increased resistance to thermal

desorption or decomposition (i.e. large change in $T_{median}$) even if the particles lost only a small volume fraction due to the isothermal evaporation (i.e. small change in VFR). We note that these composition changes are not clearly visible in the average O:C or OS$c$ values (see Table 1) and we will elaborate on possible reactions in section 3.4.

### 3.3    Residual particle composition during evaporation under dry and humid conditions

To examine changes in particle composition along the isothermal evaporation at dry conditions, we show the difference

between the normalised integrated FIGAERO-CIMS mass spectra measured at the beginning (fresh) and after 3 to 4 h of isothermal evaporation (RTC) in Figure 3 (panels (a) and (c) for the low- and high-O:C case, respectively). To investigate the changes in composition due to the humidification, panels (b) and (d) in Figure 3 show the differences between FIGAERO-CIMS mass spectra measured at dry and RH80% conditions in the beginning (fresh SOA) of the isothermal evaporation. As the low-O:C SOA particles evaporate (Figure 3a), a clear decrease in the fractional contribution of low-MW compounds

(< 300 Da, ~monomers) is observed, whereas the contribution of compounds with MW > 300 Da (predominantly dimers) increases. Correspondingly, the contribution of compounds with C > 10 increase with evaporation while that of C < 7 decreases. The relative contributions of intermediate masses are more likely to increase during evaporation, if they contain more oxygen atoms (Figure S5a). It is not possible to differentiate if these C7 - C9 compounds are really remaining in the particles or if they are simply thermal decomposition products of the more abundant dimers. However, this suggests that lighter

and/or less oxidised molecules are indeed lost more readily during isothermal evaporation in the RTC, than the heavier dimers and more oxidised compounds, which are expected to have very low vapour pressure (Mohr et al., 2017). The more detailed investigation of changes in the mass spectra (Figure S5a & b, and c & d) shows some indications of particle phase water driven chemical transformation both for low- and medium-O:C , but the differences are not as clear as in the high-O:C case (Fig. S5e and f). It should be noted that in the low-O:C case the molecules affected by particle phase water account for approximately

10% of the total signal. Therefore overall, the enhanced evaporation during FIGAERO filter collection under wet conditions is very similar to the evaporation happening under dry conditions in the RTC, and the water-driven chemistry plays only a minor role in low-O:C case. This points towards particulate water mainly reducing the viscosity and thus accelerating the mass transport in the particles as described in Yli-Juuti et al. (2017). In the high-O:C case (Figure 3c), there is also a relative decrease in masses < 300 Da with isothermal evaporation under dry conditions, but the overall picture is less clear, consistent

with very little changes in VFR and in the sum thermogram shape in this case (high-O:C, dry; Figure 1c and f). Conversely, humidifying fresh high-O:C SOA particles leads to an increase in masses < 200 Da (Figure 3d), which is a very different behaviour compared to the low-O:C cases (Figure 3a and b) or to the isothermal evaporation of high-O:C SOA particles at dry conditions (Figure 3c). Again, this suggests changes in particle composition upon humidification in the high -O:C case. The





mass fraction of compounds showing water-driven chemical transformation makes up approximately 30% of the signal in high-O:C case. This should be taken into account when process level modelling of systems representative to high O:C case is considered.

To gain a better understanding of these compositional changes related to humidification, we more closely examined the individual desorption thermograms of a few major ions (Figure 4): $C_4H_3O_6^-$ (a), $C_5H_5O_6^-$ (b), $[C_{10}H_{14}O_6+I]^-$ (c), which show an increase when the particles are humidified, and $[C_{10}H_{16}O_7+I]^-$ (d), which exhibits a net decrease in the high-O:C RH80% case. In the low-O:C case, only small shifts (0 – 6 °C) of $T_{max}$ are observed for all four ions when RH is increased. This and the changes in the thermogram shape are consistent with the behaviour observed for the total thermograms, described and explained above. In the high-O:C case, only a small shift in $T_{max}$ is visible for $[C_{10}H_{16}O_7+I]^-$ as well (Figure 4d), but for the other ions (Figure 4a-c), we see a clear shift of the thermograms, with $T_{max}$ increasing for the humidified case from 63 °C to 84 °C, 67 °C to 84 °C, and 70.5 °C to 95.5 °C for $C_4H_3O_5^-$, $C_5H_5O_6^-$, and $[C_{10}H_{14}O_6+I]^-$, respectively. This behaviour is unique for the high-O:C case. The collected organic mass loading on the FIGAERO filter was comparable (within 20%) for dry and RH80% conditions. Thus, the apparent shift of $T_{max}$ for $[C_{10}H_{16}O_7+I]^-$ (Figure 4d) could be explained by volatile material (with lower $T_{max}$) leaving the particles during evaporation in the same way as in the low-O:C case. But for the ions in Figure 4a – c, new material with higher $T_{max}$ has clearly been formed. This must be triggered by the presence of larger amounts of water under RH80% conditions, via either of two scenarios: a) there are two different isomers dominating the thermogram for the respective composition at dry vs. RH80% conditions and the desorption temperatures of these isomers differ considerably, b) the individual thermograms are dominated by the evaporation of monomers at dry conditions (or the decomposition of relatively unstable larger compounds into the observed compositions) but by thermal decomposition of (more stable) larger compounds at RH80% conditions. Isomers with higher $T_{max}$ may be formed and at the same time the isomers with lower $T_{max}$ are lost, either through reactions or through more rapid evaporation/decomposition than in the dry case. Alternatively, other low volatility material is formed that thermally decomposes into the observed compounds. With this data set, we cannot exclude either of these two explanations, but the very broad shape of the RH80% thermogram for $[C_{10}H_{14}O_6+I]^-$ at high temperatures (see Figure 4c) is an indication that at least for this ion the mechanism including thermal decomposition is more likely.

### 3.4 Possible aqueous phase processes

The FIGAERO-CIMS combined composition and thermogram measurements provided insights into the chemical composition of the residual SOA particles after humidification and/or evaporation. The observed changes in the high-O:C RH80% case can only be explained by the formation of low volatility compounds in the particle phase and removal of the corresponding higher volatility compound at the same elemental composition. Thus, we briefly consider possible processes that can explain formation of these less volatile and/or thermally more stable compounds in highly oxidised SOA at RH80% in the sections below.





Liquid water can have several effects on particle chemical composition. First, water may initiate hydration and hydrolysis reactions. Second, water may catalyse reactions between organics (e.g. Dong et al., 2018; Kaur and Vikas, 2017). Third, water reduces SOA viscosity (Hosny et al., 2016; Renbaum-Wolff et al., 2013), thereby reducing diffusional limitations to particle-phase reactions. Whereas hydrolysis generally reduces the average molecular weight of the reactants, other processes in principle enable formation of higher-MW, but thermally labile products. Water may also enhance the uptake of $O_3$ from the gas phase (Berkemeier et al., 2016; Gallimore et al., 2011), but as the average O:C ratio did not change with increased RH, any oxidising reaction can be excluded.

### 3.4.1 Homolysis of peroxy bonds (O-O) in organic peroxides (ROOH)

Organic hydro peroxides (ROOH) species are known to occur in SOA (Sanchez and Myers, 2000) and have been observed to decompose in the particle phase on timescales < 1 h under dark and wet (~40% RH) conditions (Krapf et al., 2016). Liquid-water-induced homolysis of the O-O bond in ROOH yields alkoxy (RO•) radicals that may initiate oligomerization reactions (Tong et al. 2016). Under dry conditions organic (hydro-)peroxides (ROOH) may be more stable and persist in the particle phase but decompose upon desorption and be detected as RO or R fragments in FIGAERO-CIMS. The oligomers (RO-OR) formed in the RH80% case will most likely have a lower volatility than the reactants, but they may still be prone to decompose upon desorption due to their relatively weak O-O bond. Thus, they would also be detected as RO or R fragments but at a higher apparent desorption T than the monomers.

### 3.4.2 Addition/accretion reactions

As generally no significant change in the O:C ratio was observed, only non-oxidative oligomerisation ("accretion") reactions such as aldol addition (aldehyde + aldehyde/ketone, (hemi)acetal formation (aldehyde/ketone + alcohol), peroxy(hemi)acetal formation (aldehyde/ketone + peroxide) and esterification (alcohol + (per)acid) are likely (Herrmann et al., 2015; Kroll and Seinfeld, 2008). All these reactions have reaction pathways which are enhanced by the presence of an acid catalyst in the aqueous phase. Thus, the complete lack of water in dry particles may sufficiently prevent these reactions preserving the monomer reactants. To explain our results, there would have to be an additional factor in the high-O:C case, e.g. a much higher fraction of organic peroxides and peroxy acids that form peroxy hemiacetals instead of the more stable hemiacetals. These peroxy hemiacetals would then most likely decompose at a higher temperature than the desorption temperature of the reactants, but be detected as low-MW compounds, while hemiacetals may be stable enough to be detected without decomposition.

### 4 Summary and Conclusions

We present the first study linking the oxidative age of α-pinene SOA, quantified by the O:C ratio, and isothermal evaporation, for a wide range of RH (< 2% to 80%). By utilising an RTC at room temperature and FIGAERO-CIMS as a thermal desorption technique, we were able to determine SOA volatility independent of artefacts due to thermal decomposition. At the same time, the thermal desorption data gave insights into the possible particle-phase chemistry during evaporation especially under wet



conditions. It has to be kept in mind though that the particles measured with FIGAERO-CIMS are always the residual particles after minutes to hours of isothermal evaporation either during filter collection or in the RTC.

We found a strong correlation between increased oxidation level of the initial particles and lower particle volatility expressed by less isothermal evaporation and higher $T_{max}$ values in FIGAERO-CIMS thermograms. This suggests that atmospheric
particles become more resistant to evaporation as they age over time, possibly increasing their lifetime in the atmosphere. This also means that the oxidation level needs to be kept in mind when investigating aerosol volatility in chamber or flow tube experiments. For example, for deriving VBS distributions from smog chamber yield experiments, care has to be taken that the oxidation level stays in the same range for all SOA mass loadings.

Increasing RH enhances particle evaporation as described in previous studies (Wilson et al., 2015; Yli-Juuti et al., 2017) while
the $T_{max}$ values of the residual particles were also increased. The observed changes under wet conditions in the low- and medium-O:C cases could be explained with the lowering of the particle phase viscosity alone, but there were some indications for water induced changes in the chemical composition. However, the compounds exhibiting these changes accounted for approximately 10% of the detected mass, and thus these changes are minor compared to the shift in composition due to evaporation. In the high-O:C case, strong evidence for aqueous phase reactions were found with approx. 30% of the mass
being affected. Further evidence for different processes happening in the high-O:C case was found in the relationship between the isothermal evaporation (as VFR) and thermal desorption (as $T_{median}$). It was similar for the low- and medium-O:C cases and independent of the RH. For the high-O:C case, VFR changed very little during the isothermal evaporation at RH80% while a large increase in $T_{median}$ was observed. We attribute this different behaviour to the overall different chemical composition and most likely much higher concentration of organic peroxides (ROOH) in the high-O:C case. We hypothesize
that water-induced (1) homo- and heterolytic breaking of weaker O-O bonds present in ROOH and/or (2) formation of peroxy hemiacetals may form thermally labile oligomers with enhanced yields in high-O:C SOA at RH80%. To further verify this explanation, direct measurements of the organic peroxide concentration for the different O:C cases would be needed but were not part of this study.

Our data suggests that the degree of thermal decomposition in FIGAERO-CIMS and its impact on derived volatility most
likely depends on the initial composition of the SOA and may be changed by the presence of particulate water. This highlights the benefit of isothermal methods for studying SOA particle volatility.

The SOA particles studied here had O:C ratios comparable with atmospheric SOA (typically in the 0.3 – 1.0 range, e.g., Aiken et al., 2008; Ng et al., 2010; Ortega et al., 2013), but especially the high-O:C case was probably not fully representative of atmospheric SOA as extremely high $O_3$ and OH radical exposure levels were applied in the PAM reactor. This may have led
to a larger degree of fragmentation of the precursor molecules than expected in the atmosphere, i.e., the particle phase is dominated by $C_{4-6}$ compounds instead of the expected $C_{8-10}$ compounds and their oligomers (Lambe et al., 2012). Some of the suggested aqueous phase reactions may be more likely for short carbon chain compounds (e.g. glyoxal like chemistry). Also, we can only speculate that there was a larger fraction of peroxy compounds in the particles in the high-O:C case as we had no direct measurement. However, recent studies with ambient SOA have shown that these particles can contain large amounts of



environmental persistent radicals and are prone to form C- and O-centred organic radicals when wetted, which can start oligomerisation reactions in the particle phase (Arangio et al., 2016). So, although we formed the high-$\overline{O:C}$ SOA in a PAM reactor under extreme conditions, the produced particle and their behaviour allowed us to study processes which are most likely atmospherically relevant. Hence, our study substantially increases the understanding of complicated and inadequately

5 studied particle phase processes and the results highlight the importance of water-driven chemistry in SOA.

## 5 Author contribution

AV, TY-J, and AB designed the study, AB, ATL, AY, ZL, CF, EK, LH, CM, and SN performed the measurements, AB, AY, ZL, EL, LH, WH, CM, DR, SN, TY-J, SS, and AV participated in data analysis and/or interpretation, O-PT., OL, and TY-J. performed the model calculations, AB, AV, and SS wrote the manuscript.

## 6 Acknowledgements

We thank the European Research Council (ERC StG QAPPA 335478), the Academy of Finland Centre of Excellence program (decision 307331) and the Academy of Finland (grants 299544, 317373 and 310682) for financial support. SN acknowledges the Fulbright Finland Foundation and the Saastamoinen Foundation that funded his visit to the University of Eastern Finland. ATL acknowledges support from the Atmospheric Chemistry Program of the US National Science Foundation under grant no.

15 AGS-1537446.

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





## 8    Figures



**Figure 1: Evapograms (left column: a-c) and total thermograms (right column: d-f) for low-Ơ:C (top row), medium-Ơ:C (middle row) and high-Ơ:C (bottom row). Coloured boxes in evapograms indicate FIGAERO sampling time. Thermograms are normalised with total signal area.**



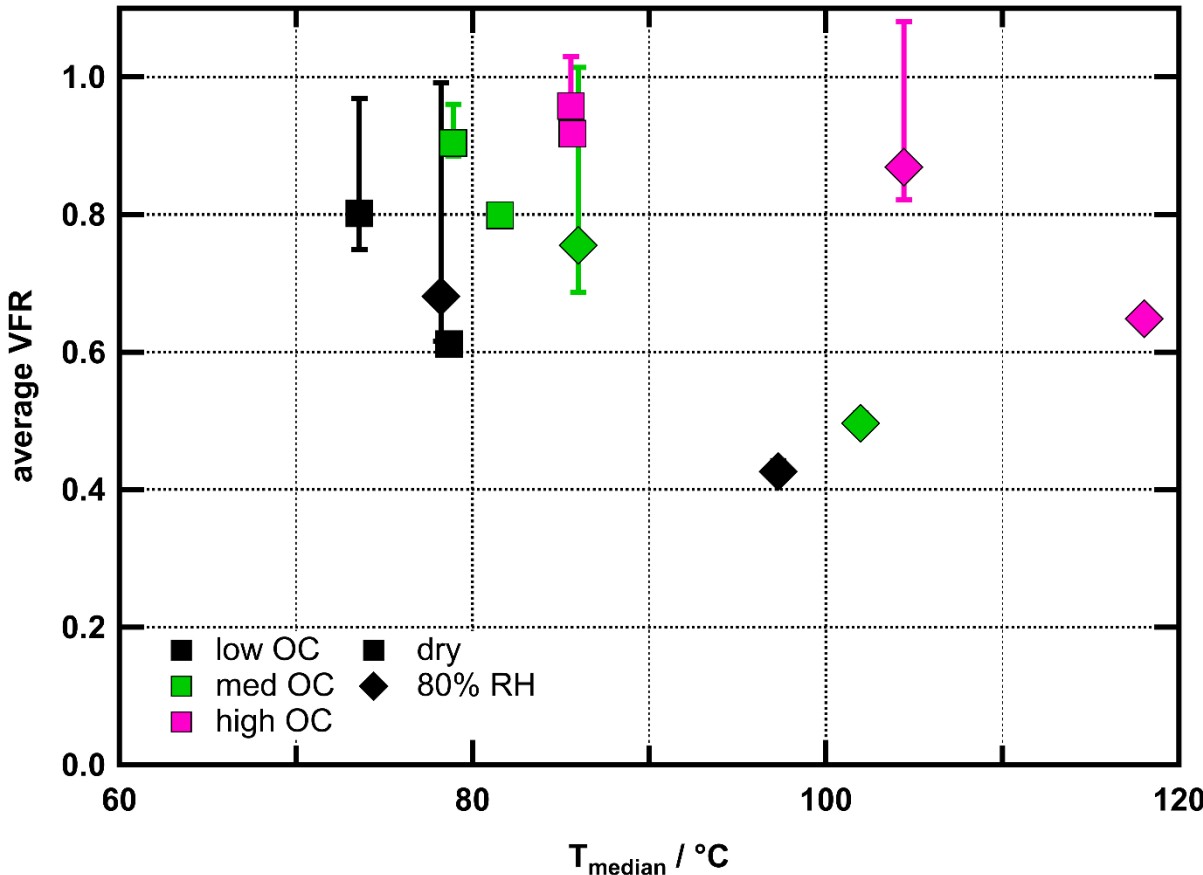

**Figure 2: Average VFR during FIGAERO sampling vs median desorption temperature (T$_{median}$) for all experiments. Colours indicate O:C ratios. Measurements under dry conditions are marked with squares, those under RH80% with diamonds. Error bars indicate minimum and maximum VFR observed during sampling time.**







**Figure 3: Changes in normalised spectra for low-O̅:C̅ (top panels: a, b) and high-O̅:C̅ (bottom panels: c, d). Left column (a, c): changes due to evaporation under dry conditions, right column (b, d): changes between dry and wet conditions. Colour indicates number of C atoms in the identified ions (black corresponds to C-number of larger than 10). Mass spectra were normalised by total signal and then the difference calculated.**





**Figure 4: Not-normalised thermograms for four single ions for low- and high-O̅:C cases. Ions in panels (a), (b), and (c) show a net increase in the RH80% case while the ion in panel (d) had a net decrease in the high-O̅:C ratio case. Note that the amount of SOA mass collected on the FIGAERO filter was 5 – 20% higher in the RH80% cases and is also different between the O̅:C cases.**

