# Peer review of "Insights into the O:C dependent mechanisms controlling the evaporation of $\alpha$ -pinene secondary organic aerosol particles"

_Atmospheric Chemistry and Physics, 2018_

## Referee Comment (RC1) · Anonymous Referee #1 · 31 Jan 2019

This manuscript describes results from experiments on isothermal evaporation of a-pinene SOA of different oxidation state and at different relative humidity. The authors observed reduced evaporation at higher oxidation state, consistent with an inverse relationship between organic aerosol volatility and oxidation state. They observed increased evaporation at higher relative humidity, potentially due to changes in particle viscosity and aqueous chemistry. The experiments are well designed, the data analysis is solid, and the results, implications and limitations are well described. In my opinion this manuscript can be published as is.

[Figure]

2019.

---

## Referee Comment (RC2) · Anonymous Referee #2 · 21 Feb 2019

Buchholz et al propose a study investigating the mechanisms controlling the evaporation of biogenic-derived secondary organic aerosol (SOA) formed from the photooxidation of alpha-pinene. Two mass spectrometers were used to retrieve the chemical composition of the SOA throughout the evaporation processes. The size distribution of the particle was also characterized. Overall this study is very well constrained and the results well presented. However, I have a few comments/concerns, mainly regarding the FIGAERO data, that should be at least discussed in the manuscript.

My main concern is the absence of blank measurements with the FIGAERO. As it has been initially discussed by Lopez-Hilfiker et al. and later in other studies, performing

blank measurements is crucial to validate the chemical information obtained using a FIGAERO. The authors should discuss this point and explain how they made sure that their results were not impacted by the background of the instrument. To me, this discussion is important since the mass collected onto the filters was particularly low. In addition the authors should clarify the following points: Was the FIGAERO sampling the gas phase coming from the lab, the PAM chamber or clean air during the aerosol sampling time? This is important to know as it can significantly impact the background of the instrument.

page 6, lines 15-17: Some information regarding the declustering strength should be provided (e.g., 173/127). Why did the authors make this choice rather than reducing the declustering and be more sensitive to a wider variety of compounds?

page 7, lines 1-5: If the gas phase was sampled during the PAM chamber experiments it should have been possible to observe the larger formation of small carbonyls. Please clarify.

page 7, lines 5-10: How do the thermogram look like for those small compounds? Are the thermograms consistent with SVOC or (E)LVOC?

page 7, lines 15-20: The observations made by the authors are not consistent with a recent modeling study performed by DeRieux et al (https://doi.org/10.5194/acp-18-6331-2018) and not well constrained. The authors mentioned that the slower evaporation under dry conditions is due to diffusion limitation. While the LWC and alpha-pinene-derived SOA viscosity at 40% and 80% can be anticipated to be significantly different (i.e., at RH 0% 10E+8Pa s, 40% 10E+6 Pa s and 80% 10E+2) the evaporation rates are similar. That's confusing and it should be discussed and the results better constrained (estimation of the LWC, viscosity,...) in the paper.

page 8, lines 3-7:How do the O:C ratios evolve as a function of evaporation? Does it increase or decrease? The authors should be able to track these changes.

page 8, lines 17-20: Could it be possible that sampling wet air (i.e., 80%) leads to larger adsorption of gaseous compounds onto the filter (i.e., positive artifact)?

page 9, lines 18-20: The authors should compare some of their results with the recent work published by Riva et al. 2019 (doi.org/10.1038/s41612-018-0058-0). In this earlier study, the authors have shown that particle phase processes lead to the formation of oligomers that further decompose into C7-C9 compounds. How do the average DBE value change?
* * *

---

## Author Comment (AC1) · 21 Mar 2019

We want to thank the reviewer for carefully reading our manuscript and providing such a positive review.

---

## Author Response (AR1)

**Response to reviewer #2**

We thank the reviewer for the careful reading of our manuscript. Below we address each of the reviewer's comments and indicate the requested changes to the manuscript. The reviewer's comments are marked in blue, our response in black, and the changes to the manuscript/SI material text are indicated in red.

Buchholz et al propose a study investigating the mechanisms controlling the evaporation of biogenic-derived secondary organic aerosol (SOA) formed from the photooxidation of alpha-pinene. Two mass spectrometers were used to retrieve the chemical composition of the SOA throughout the evaporation processes. The size distribution of the particle was also characterized. Overall this study is very well constrained and the results well presented. However, I have a few comments/concerns, mainly regarding the FIGAERO data, that should be at least discussed in the manuscript.

**Comment 1**

My main concern is the absence of blank measurements with the FIGAERO. As it has been initially discussed by Lopez-Hilfiker et al. and later in other studies, performing blank measurements is crucial to validate the chemical information obtained using a FIGAERO. The authors should discuss this point and explain how they made sure that their results were not impacted by the background of the instrument. To me, this discussion is important since the mass collected onto the filters was particularly low. In addition the authors should clarify the following points: Was the FIGAERO sampling the gas phase coming from the lab, the PAM chamber or clean air during the aerosol sampling time? This is important to know as it can significantly impact the background of the instrument.

**Response**

During particle collection, the gas phase inlet of CIMS was connected to purified compressed air. We made sure that after sampling high concentrations (i.e., measuring directly from PAM) the signal had returned to background values before a thermogram was measured. The gas phase inlet of CIMS was capped overnight, and the particle phase inlet was without sampling flow open to ambient air via the sampling line (stainless-steel and Tygon® silicon tubing, 4 mm inner diameter).

We agree that blank filter measurements are indeed important to fully validate the chemical composition information from FIGAERO-CIMS measurements. However, for the analysis and interpretation we present in our paper, the filter and instrument backgrounds were small compared to particle-phase signals (up to 180 °C desorption temperature) and had little impact on the resulting thermograms, and none on the overall conclusions. This is because most of the background signal appeared at desorption temperatures above 180 °C and showed a steady increase with temperature while the prominent ions in the difference spectra appeared at lower desorption temperature and had distinct peak shapes. To illustrate the impact of subtracting

the first blank of the day (i.e., the upper limit of instrument background) on the analysis, we calculated the difference spectra for the high-O̅:C case after this subtraction (Figure R 3a and b) and compared them with those calculated from the data with no background removal (Figure R 3c and d and Figure 3c and d in the manuscript). The values in the difference spectra changed due to background removal by less than $1 \cdot 10^{-4}$ for 90% of the identified ions. The pattern of increase/decrease due to evaporation or humidification did not change. Instead the magnitude of change (e.g. an ion has an increased contribution after humidification) is even larger when the background is removed. Thus, the overall interpretation of the data is not impacted by accounting for the filter background measurements. Due to the varying availability and quality of the blank measurements, we decided to keep the data without background removal in the manuscript.

We will include the following paragraphs and figures discussing the filter blank measurements in the SI material and make a reference to it in the main text (page 5 line 30):

The raw data was averaged to provide average mass spectra spaced by 20 s, and baseline correction was applied before fitting the high-resolution mass spectral data. Details about the magnitude and impact of the instrument background (filter blank measurements) on our analysis are discussed in the SI material (Section 1.2).

**1.2 FIGAERO-CIMS instrument background**

One to two filter blank measurements were performed in the morning of each experiment day. To illustrate the results of the blank measurements, we show the non-normalised integrated mass spectra for all conducted blank measurements for the low-O̅:C case in Figure S6a and b. The non-normalised total thermograms (Figure S6c) clearly show that even for the lowest collected mass (RH80%, RTC case, light blue line) the total signal is still much higher overall than the corresponding blank measurements (purple and pink lines in Figure S6c). Another example is presented in Figure S7: the non-normalised integrated mass spectra for the high-O̅:C cases. On that day, only one blank measurement was performed. It is apparent that a few ions are clearly elevated in this blank measurement, but generally the ion abundances observed during measurements are much higher than those in the background spectra.

We have categorised background signals in the FIGAERO-CIMS measurements into two types: 1) compounds being emitted from the filter/set-up during the desorption, especially at the highest desorption temperatures, and 2) compounds accumulated on the filter from ambient air while in "idle" position (no flow through filter but inlet open to room air). Type 1 compounds should be relatively constant throughout an experiment day, but the abundance of type 2 compounds will depend on how long the filter has been in the idle position and will be removed with each heating cycle (including the 1-2 initial blank measurements). The first filter blank measurement in the morning was conducted after 10 - 14 h of idle time overnight with the second blank following within a few minutes after the first one. During the following experiments of the day, there were typically 1 - 2 h between the end of desorption of one sample and the collection of the next. Thus, the first blank should be considered as an upper limit of contamination/background (both with type 1 and 2 compounds) while the second one may be seen as the lower limit for type 1 compounds (and a second measure of the upper limit for type 2 compounds). As the lengths of idle times were so different, the morning filter blank measurements are not fully representative of the situation throughout

the day. Therefore, subtracting the available blank measurements from the corresponding experiment data was deemed to be impractical, especially in those cases where only one blank measurement was available, as for some ion signals the blank subtraction would lead to negative signal values, which are unphysical. However, we carefully compared the difference spectra for uncorrected data (Figure 3c and d) and data which had the maximum background subtracted (panels a and b). When removing the estimated upper limit of instrument background, the overall patterns in the difference spectra stay the same. 90% of the ions exhibit a change of less than $1 \cdot 10^{-4}$ for the values in the difference spectra. For some ions, the increases/decreases due to humidification/evaporation become even more prominent. This finding combined with the fact that the quality and availability of blank measurements varied between SOA types, we decided to show the uncorrected difference spectra in the main manuscript depicting the minimum change to be expected due to humidification and/or evaporation.

**Comment 2**

page 6, lines 15-17: Some information regarding the declustering strength should be provided (e.g., 173/127). Why did the authors make this choice rather than reducing the declustering and be more sensitive to a wider variety of compounds?

**Response**

We had switched the ionisation of the CIMS from acetate to iodide just before these measurements. Using acetate as reagent ions, we had found it most useful to use fairly strongly declustering settings as has become common practice for that technique (Brophy and Farmer, 2016). When we made the switch to iodide as reagent ion, unfortunately we did not change the voltage settings to lower declustering strength (as is typical practice for iodide-CIMS). This oversight only became apparent when we conducted the detailed analysis of the high-resolution data after the measurements had finished.

We are not aware of general marker compounds for a measure of declustering strength for iodide CIMS, and we have been unable to conclude what method the reviewer had in mind when referring to the ratio of 173 (m/z of formic acid iodide cluster) to 127 (m/z of iodide). The ratio of analyte-iodide clusters and declustered signals (assuming deprotonation) strongly depends on the individual ion. There are even examples, where for one identified sum formula pair (e.g., $C_4H_5O_4^-$ and $[C_4H_6O_4 \; I]^-$, see Figure R 4 (a)) the ratio changes between low-, med- and high-$\overline{O:C}$ experiments. This suggests that there may be more complex declustering and/or fragmentation reactions taking place depending on the molecular structure of the parent molecule. This makes it near impossible to derive a general measure of declustering strength. We could however list the voltage settings of the CIMS ion guidance elements in the SI material if the reviewer insists.

The reviewer is correct in implying that our settings not only made our CIMS measurements more strongly declustering than necessary, but also less sensitive overall than necessary. However, the scientific results of our study do not rely on high or accurate (quantitative) sensitivity overall. And any signal that we may have lost in noise due to potentially low sensitivity would by default be small contributors to the CIMS mass spectra and thermograms.

**Comment 3**

page 7, lines 1-5: If the gas phase was sampled during the PAM chamber experiments it should have been possible to observe the larger formation of small carbonyls. Please clarify.

**Response**

As we did not use an $O_3$ scrubber for the gas phase measurements with CIMS, the ppm level $O_3$ concentrations coming from the PAM severely depleted the primary ion ($I^-$), making the analysis of the gas phase data challenging. As the $O_3$ concentrations were 6.4, 16, and 22 ppm for low-, medium-, and high-O:C settings, comparison of signal strength of individual ions between the different settings are thus problematic for the gas-phase data analysis. However, we want to emphasize that all FIGAERO-CIMS measurements (particle-phase) presented in the manuscript were conducted with monodisperse samples, i.e., samples were collected after the dilution of the gas phase in the NanoDMAs. Hence, the problems caused by high $O_3$ concentrations had no effect on the results and analysis presented in the manuscript.

**Comment 4**

page 7, lines 5-10: How do the thermogram look like for those small compounds? Are the thermograms consistent with SVOC or (E)LVOC?

**Response**

We plotted the thermograms of four typical low $M_w$ ions in Figure R 4. They all show distinct peaks typically associated with single (or few, similar) evaporating compounds (in contrast to the wide tails at high temperatures usually expected for ongoing thermal decomposition from multiple sources). Only one example shows an increase at high desorption temperatures (Figure R 4 (d)) indicating that this part of the signal stems most likely from thermal decomposition of larger compounds. But this behaviour is the exception and not the rule.

For any single compound's thermogram, the temperature of peak desorption ($T_{max}$) relates to its volatility. We calibrated the connection between $p_{sat}$ (and C*) and FIGAERO temperature of maximum desorption similar to Bannan et al. (2019), and indicate that relationship by coloured areas corresponding to volatility classes defined by the $C_{sat}^*$ ranges in Donahue et al. (2012) in Figure R 4. With this classification, these low $M_w$ ions have $T_{max}$ values in the range of SVOC and LVOC (Table R1). Using the parameterisation from Li et al. (2016), we calculated the $\log10(C_{sat}^*)$ from the elemental composition for these ions (see Table R1). These values would categorise three of them as IVOCs ($\log10(C_{sat}^*) > 2$) and $C_4H_4O_6$ as SVOC. Even taking into account the uncertainty of such parameterisations, the difference between $\log10(C_{sat}^*)$ values from composition parameterisation and $T_{max}$ measurements is large and indicates that these are indeed thermal decomposition products despite their narrow peak shape (the most likely exception being $C_4H_4O_6$).

**Comment 5**

page 7, lines 15-20: The observations made by the authors are not consistent with a recent modeling study performed by DeRieux et al (https://doi.org/10.5194/acp-18-6331-2018) and not well constrained. The authors mentioned that the slower evaporation under dry conditions is due to diffusion limitation. While the LWC and alpha-pinene-derived SOA viscosity at 40% and 80% can be anticipated to be significantly different (i.e., at RH 0% 10E+8Pa s, 40% 10E+6 Pa s and 80% 10E+2) the evaporation rates are similar. That's confusing and it should be discussed and the results better constrained (estimation of the LWC, viscosity,...) in the paper.

**Response**

Our results are not inconsistent with the study by DeRieux et al. (2018), as our results don't imply that the viscosity at RH40% and RH80% would be the same. As discussed in Yli-Juuti et al., (2017), lower diffusivity at RH40% compared to RH80% does not necessarily manifest in slower evaporation rates. As long as the viscosity is low enough - for our system below approximately $10^5$ Pa s (Yli-Juuti et al., 2017) - the limiting process is evaporation of the molecules from the particle surface, i.e. the vapour pressure of the molecules and gas phase equilibration time scales. So our results do not disagree with the notion of an increasing viscosity with decreasing RH, as shown e.g. by DeRieux et al. (2018), but rather suggest that at RH40% the viscosity is still low enough for particle phase diffusion not to significantly limit the evaporation, while at the dry conditions the particle phase diffusivity is a limiting process.

When the evaporation curves are investigated in detailed, it can be seen that the evaporation is actually a bit faster at RH40% than at RH80%. This is due to the Raoult effect (i.e. higher LWC at higher RH), as discussed in Yli-Juuti et al. (2017), and also the SI of this manuscript. This effect of particles being more diluted at higher RH is included in our modelling.

Yli-Juuti et al. (2017) showed that the observed evaporation behaviour under dry conditions could only be reproduced if the particle viscosity was allowed to change with evaporation, i.e. with particle composition. As this is the case also for the experiments in this study, our data indicates that the viscosity of $\alpha$-pinene SOA strongly depends on the exact chemical composition of the particles (e.g. how much semi-volatile compounds are left) which is also in consistent with DeRieux et al. (2018).

As the focus of this paper is not on the modelling results, we will add the following paragraph to the description of the modelling in the SI material and add the KM-GAP model evapogram curves to Figure S4.

To validate our assumption of liquid like behaviour at RH40%, we calculated the evaporation curve at RH40% using the starting VBS distribution derived from the RH80% case both with the LLEVAP (assuming liquid-like behaviour) and with the KM-GAP model (assuming mass transfer limitations, applying viscosity derived from dry case). The curves are shown in Figure S4. In the medium- and high-O:C cases, the LLEVAP curve (dashed line) clearly represents the measured data points. In the low-O:C case, LLEVAP represents the early stages of evaporation better while the later part is closer to the KM-GAP

**Comment 6**

page 8, lines 3-7: How do the O:C ratios evolve as a function of evaporation? Does it increase or decrease? The authors should be able to track these changes.

**Response**

Generally, the mass concentration was very low ($<0.5$ µg/m$^3$) for the long evaporation measurements, and thus the quality of AMS O:C ratio data was very poor. Longer measurement times were not an option as that would have increased the necessary dilution and depleted the particle concentration in the RTC even faster. For some of the high concentration experiments (for FIGAERO sampling), mass loadings were high enough to derive reliable O:C ratios (Table R1: Molecular mass, $T_{max}$ and $\log10(C^*_{sat})$ derived from composition parameterisation (Li et al., 2016) and calibration of peak desorption temperature. $T_{max}$ values are for the medium-case.

| sum formula | $M_w$ g mol$^{-1}$ | $\log10(C^*_{sat})$ composition | $T_{max}$ °C | $\log10(C^*_{sat})$ $T_{max}$ |
|---|---|---|---|---|
| C$_4$H$_6$O$_4$ | 118.03 | 4.85 | 72 | -0.7 |
| C$_3$H$_4$O$_4$ | 104.01 | 4.41 | 44 | 0.8 |
| C$_4$H$_4$O$_6$ | 148.00 | 1.26 | 60 | -0.1 |
| C$_2$H$_4$O$_2$ | 60.02 | 7.5 | 52 | 0.4 |

Table R2). In those cases, no clear trends were visible for changes in O:C values with evaporation or humidification (changes of less than 0.08).

Average O:C ratios from FIGAERO measurements are given in Table 1 in the manuscript. Again, no clear trend was observed with evaporation. This is mentioned later in the manuscript (page 9, lines 6-7). But we will strengthen that statement in the main text at page 9 line 9.

During the evaporation the initial O:C changed very little (Table 1). This is consistent with earlier observations reported by Yli-Juuti et al. (2017) who interpreted this as evidence for the presence of low volatility oligomers in the particles. These should have very similar O:C ratios to the corresponding monomers.

**Comment 7**

page 8, lines 17-20: Could it be possible that sampling wet air (i.e., 80%) leads to larger adsorption of gaseous compounds onto the filter (i.e., positive artifact)?

**Response**

In principle, it is true that the adsorption of gaseous compounds may be enhanced under wet conditions. However, in our study the concentrations of gaseous compounds were very low as we were not sampling the aerosol (gas and particle phase). Instead, we separated the particles from the original gas phase during the size selection in the NanoDMAs. During evaporation in the RTC, the stainless-steel walls present a large enough sink for the evaporating vapours so that there is no build-up in the gas phase for the RTC samples. We showed in previous characterisation experiments for the study reported in Yli-Juuti et al. (2017) that even an increase of particle concentration by an order of magnitude did not change the evaporation behaviour, which is a clear sign that the instantaneous wall loss assumption for evaporating vapours is reasonable.

**Comment 8**

page 9, lines 18-20: The authors should compare some of their results with the recent work published by Riva et al. 2019 (doi.org/10.1038/s41612-018-0058-0). In this earlier study, the authors have shown that particle phase processes lead to the formation of oligomers that further decompose into C7-C9 compounds. How do the average DBE value change?

**Response**

We carefully studied the work by Riva et al. (2019). They show very interesting findings about $\alpha$-pinene SOA produced from ozonolysis reactions with and without seed present. Unfortunately, they do not state $O:C$ (or $\overline{OS}_C$ ) values for their SOA. As we show in our study, the initial $O:C$ ratio has a large influence on the composition, evaporation and thermal desorption behaviour. Also, all our SOA was produced via a combination of photooxidation and ozonolysis without any seed.

Nonetheless, we calculated the DBE changes as the reviewer suggested and depict them in Figure R 6 in the same fashion as in Supplement Figure 8 in Riva et al. (2019). In the low-$O:C$ case, the C7-9 compounds with increased contribution after evaporation have higher DBE values (green colours in Figure R 6a), but no such clear trend can be found for medium- or high-$O:C$ SOA particles (panels c and e). When comparing high-$O:C$ dry and wet particles, we see a shift towards lower DBE values in the wet case. This again highlights that different processes occur in the high-$O:C$ case when particles are wetted.

We would like to point out that Riva et al. (2019) observed the strongest increase of "other oligomeric" compounds (HMW, C7-C10 compounds with high DBE) in the dry ABS seed case, when $HSO_4^-$ and $H^+$ ions are available in the particles. It is known that under these conditions organosulphates (i.e. sulphate esters) can be formed (Liggio and Li, 2006; Surratt et al., 2007). Dimers with a sulphate ester link are also possible. This group of compounds is known to be thermally instable (Hu et al., 2011) and thus would mostly be detected as the corresponding decomposition products in FIGAERO-CIMS. But these compounds are highly unlikely to form in our study as no seed was used and no significant amounts of sulphate was detected in the AMS.

We will cite Riva et al. (2019) in the manuscript text (page 12, line 27) as an example for particle phase processes that can form low volatile but thermally instable compounds. However, we believe that different chemical processes are at work here, and thus the resulting chemical compounds are most likely also very different.

Our data suggests that the degree of thermal decomposition in FIGAERO-CIMS and its impact on derived volatility most likely depends on the initial composition of the SOA and may be changed by the presence of particulate water. Another recent study has shown that chemical composition changes induced by the presence of acidic inorganic seeds may also produce low volatile, but thermally instable compounds, which can only be detected as their decomposition products with FIGAERO-CIMS (Riva et al., 2019). This highlights the benefit of isothermal methods for studying SOA particle volatility.

**Tables**

Table R1: Molecular mass, $T_{max}$ and $\log10(C_{sat}^*)$ derived from composition parameterisation (Li et al., 2016) and calibration of peak desorption temperature. $T_{max}$ values are for the medium-O:C case.

| sum formula | $M_w$ g mol$^{-1}$ | $\log10(C_{sat}^*)$ composition | $T_{max}$ °C | $\log10(C_{sat}^*)$ $T_{max}$ |
|---|---|---|---|---|
| $C_4H_6O_4$ | 118.03 | 4.85 | 72 | -0.7 |
| $C_3H_4O_4$ | 104.01 | 4.41 | 44 | 0.8 |
| $C_4H_4O_6$ | 148.00 | 1.26 | 60 | -0.1 |
| $C_2H_4O_2$ | 60.02 | 7.5 | 52 | 0.4 |

Table R2: Average O:C values and mass concentration measured with AMS for monodisperse SOA. Values in *italics* indicate that the AMS mass concentration was well below 0.5 µg m$^{-3}$ and derived values should be used carefully.

| OH exposure | sampling condition | | mass / µg m$^{-3}$ | O:C |
|---|---|---|---|---|
| low | dry | fresh | 3.37 | 0.54 |
| | | RTC | *0.33* | *0.53* |
| | RH80% | fresh | 3.29 | 0.53 |
| | | RTC | *0.22* | *0.53* |
| medium | dry | fresh | 4.45 | 0.69 |
| | | RTC | 0.72 | 0.68 |
| | RH80% | fresh | 5.96 | 0.67 |
| | | RTC | 0.5 | 0.61 |
| high | dry | fresh | 2.49 | 0.96 |
| | | RTC | 0.46 | 0.92 |
| | RH80% | fresh | 3.3 | 0.97 |
| | | RTC | *0.3* | *0.89* |

**Figures**

[Figure]

**Figure R 1: Non-normalised integrated mass spectra of all filter blank measurements for low-O:C (panels (a) and (b)) and non-normalised total thermograms (c) for filter blanks and measurements after evaporation in the RTC (orange: dry, light blue: RH80%). The colour code is the same in all three panels.**

[Figure]

**Figure R 2:** Non-normalised integrated mass spectra for high-O:C cases with highest (a) and lowest (b) mass loading on the FIGAERO filter. Panel (c) shows the first filter blank measurement in the morning of the same day (i.e. maximum background).

[Figure]

**Figure R 3: Changes in normalised spectra for high-O̅:C̅ -cases with background subtraction (top panels: a, b) and no background subtraction (bottom panels: c, d). Left column (a, b): changes due to evaporation under dry conditions, right column (b, d): changes between dry and wet conditions. Colour indicates number of C atoms in the identified ions (black corresponds to C-number of larger then 10). Mass spectra were normalised by total signal and then the difference calculated. Note that panels (c) and (d) are identical to those in Figure 3 in the main manuscript.**

[Figure]

**Figure R 4: Non-normalised ion thermograms for four low $M_w$ ions for fresh SOA of all O:C cases. For each ion, thermograms are given for the I$^-$ cluster (dashed lines) and the corresponding deprotonated ion (solid lines). Background colour indicates volatility classes according to Donahue et al. (2012) and our peak desorption temperature ($T_{max}$) calibrations.**

[Figure]

**Figure R 5: Measured and modelled evapogram data. Dashed lines indicated model results using KM-GAP (assuming mass transport limitations in the particles), solid lines are results from LLEVAP (liquid like behaviour).**

[Figure]

**Figure R 6: Changes in DBE during dry evaporation (left) and humidification (right) for all O:C cases. Red colours indicate that signal was higher for the fresh dry particles.**

[revised manuscript text omitted]

**Supplement material**

**1.1    FIGAERO-CIMS declustering**

As described in section 2.2, we observed a considerable amount of declustered ions in the FIGAERO-CIMS mass spectra. We speculate that a possible declustering process was the removal of HI from $[M+I]^-$ adducts, leaving behind $[M-H]^-$ to be detected, e.g., carboxylate anions following the deprotonation of the corresponding carboxylic acids. However, it is also possible that other decomposition reactions occurred, such as decomposition of peroxyacid-iodide adducts into carboxylate anions ($[M-OH]^-$, Lee et al., 2014)), decarboxylation $[M – H – CO_2]^-$ and/or dehydration of carboxylic acids and alcohols $[M – H – H_2O]^-$ (e.g. Canagaratna et al., 2015; Stark et al., 2017), or cleavage of weak organic peroxide bonds (Iyer et al., 2016; Schobesberger et al., 2018). In general, declustered ions were observed at relatively lower average desorption temperatures. Either the respective parent compounds are, on average, more volatile than compounds observed as $[M+I]^-$, or higher desorption temperatures induce decomposition processes that forestall potential CIMS-induced decomposition. Clearly, dedicated studies are warranted to gain a mechanistic understanding of the combination of thermally induced (in the FIGAERO) and collision-induced (in the CIMS) dissociation of ion clusters and/or molecules.

**1.2    FIGAERO-CIMS instrument background**

One to two filter blank measurements were performed in the morning of each experiment day. To illustrate the results of the blank measurements, we show the non-normalised integrated mass spectra for all conducted blank measurements for the low-O:C case in Figure S6a and b. The non-normalised total thermograms (Figure S6c) clearly show that even for the lowest collected mass (RH80%, RTC case, light blue line) the total signal is still much higher overall than the corresponding blank measurements (purple and pink lines in Figure S6c). Another example is presented in Figure S7: the non-normalised integrated mass spectra for the high-O:C cases. On that day, only one blank measurement was performed. It is apparent that a few ions are clearly elevated in this blank measurement, but generally the ion abundances observed during measurements are much higher than those in the background spectra.

We have categorised background signals in the FIGAERO-CIMS measurements into two types: 1) compounds being emitted from the filter/set-up during the desorption, especially at the highest desorption temperatures, and 2) compounds accumulated on the filter from ambient air while in "idle" position (no flow through filter but inlet open to room air). Type 1 compounds should be relatively constant throughout an experiment day, but the abundance of type 2 compounds will depend on how long the filter has been in the idle position and will be removed with each heating cycle (including the 1-2 initial blank measurements). The first filter blank measurement in the morning was conducted after 10 - 14 h of idle time overnight with the second blank following within a few minutes after the first one. During the following experiments of the day, there were typically 1 - 2 h between the end of desorption of one sample and the collection of the next. Thus, the first blank should be considered as an upper limit of contamination/background (both with type 1 and 2 compounds) while the second one may be

seen as the lower limit for type 1 compounds (and a second measure of the upper limit for type 2 compounds). As the lengths of idle times were so different, the morning filter blank measurements are not fully representative of the situation throughout the day. Therefore, subtracting the available blank measurements from the corresponding experiment data was deemed to be impractical, especially in those cases where only one blank measurement was available, as for some ion signals the blank subtraction would lead to negative signal values, which are unphysical. However, we carefully compared the difference spectra for uncorrected data (Figure 3c and d) and data which had the maximum background subtracted (panels a and b). When removing the estimated upper limit of instrument background, the overall patterns in the difference spectra stay the same. 90% of the ions exhibit a change of less than $1 \cdot 10^{-4}$ for the values in the difference spectra. For some ions, the increases/decreases due to humidification/evaporation become even more prominent. This finding combined with the fact that the quality and availability of blank measurements varied between SOA types, we decided to show the uncorrected difference spectra in the main manuscript depicting the minimum change to be expected due to humidification and/or evaporation.

**1.2 1.3 FIGAERO-CIMS average values**

For the FIGAERO data, the average composition and elemental ratios were derived from the identified sum formula for each ion. The average composition (average number of C, H, and O atoms) was calculated as the signal weighted sum:

$$composition = \sum_i (C_i \cdot f_i), \sum_i (H_i \cdot f_i), \sum_i (O_i \cdot f_i) \tag{S1}$$

With $C_i$, $H_i$, and $O_i$ being the number of C, H, and O atoms in the sum formula for each ion $i$ and $f_i$ the normalised signal of the ion $i$, i.e., the count rate of ion $i$ normalising to the sum over all non-reagent ions.

For each identified sum formula, the O:C and H:C ratios were calculated. The average $\overline{O:C}$ and $\overline{H:C}$ ratios were calculated as the signal weighted sum of these:

$$\overline{O:C} = \sum_i ((O:C)_i \cdot f_i) \tag{S2}$$

$$\overline{H:C} = \sum_i ((H:C)_i \cdot f_i) \tag{S3}$$

with $(O{:}C)_i$ (or $(H{:}C)_i$) being the O:C (or H:C) ratio calculated from the sum formula of each ion $i$ and $f_i$ the normalised signal of the ion $i$. Note that this is not the same as the ratio of O and C in the average composition.

**1.3 1.4 Modelling of evaporation**

The evaporation inside the RTC was modelled with two different process models. The models were used together with an optimization algorithm to investigate if the difference in evaporation between 80% and 40% RH can be explained by the solution effect (Raoults law). In both models the particle composition was presented with a one-dimensional VBS (Donahue et al., 2006) with 6 compounds or 'bins' spanning from $10^{-3}$ µg m$^{-3}$ to $10^2$ µg m$^{-3}$ with a decade difference between two adjacent

bins. The evaporation at RH40% and RH80% was modelled with a liquid-like evaporation model (LLEVAP; Yli-Juuti et al. (2017)) where the particles are assumed to behave like well-mixed liquids. Thus, the limiting step in evaporation is the transport of mass between particle and gas phases. The evaporation under dry conditions was modelled with a slightly modified version of the kinetic multi-layer model for gas-particle interactions in aerosols and clouds (KM-GAP ,Shiraiwa et al., 2012; Yli-Juuti et al., 2017). In KM-GAP, the viscosity in each layer of the particle was expressed with a mixing rule (O'Meara et al., 2016):

$$\textbf{log}_{10}(\eta_j) = \sum_{i=1}^{N} X_{mole,i,j}\,\textbf{log}_{10}(b_i) \tag{S4}$$

where $\eta_j$ is the viscosity in the $j$th layer, $X_{mole,i,j}$ is the molar fraction of the $i$th compound in $j$th layer, and $b_i$ is a coefficient that describes how much compound $i$ affects the viscosity. The particle phase diffusion coefficients were calculated from the viscosity with the Stokes-Einstein equation. Both models assume ideal solution and calculate the water uptake based on continuous equilibrium between gas and particle phase (Yli-Juuti et al., 2017). In all simulation cases, the molar masses of each bin were set to 200 g mol$^{-1}$, particle phase densities to 1200 kg m$^{-3}$, and gas phase diffusion coefficients to 0.05 cm$^2$ s$^{-1}$. The two process models were coupled to a global optimization algorithm Monte Carlo Genetic Algorithm (MCGA, Berkemeier et al., 2017). In the optimization process, the free parameters, i.e. the parameters that the MCGA can vary, were the mole fraction of each VBS bin when the particles enter the residence time chamber and the coefficients $b_i$ in Eq. S4. The MCGA was set to seek for a set of free parameters that minimizes the mean-squared-error between the measured and simulated evapograms. For each O:C case, the parameters were optimized simultaneously to evaporation data at RH80% and dry conditions. This yields the initial particle composition in term of the VBS bins assuming that the difference between evaporation rates is controlled by the low particle phase diffusivity in dry conditions. This initial composition is expected to be the same for all humidity conditions due to the experimental procedure. The initial composition was then used in the LLEVAP to simulate evaporation at 40% RH. The resulting evapogram curves are shown in Figure S4 for all experiments.

To validate our assumption of liquid like behaviour at RH40%, we calculated the evaporation curve at RH40% using the starting VBS distribution derived from the RH80% case both with the LLEVAP (assuming liquid-like behaviour) and with the KM-GAP model (assuming mass transfer limitations, applying viscosity derived from dry case). The curves are shown in Figure S4. In the medium- and high-O:C cases, the LLEVAP curve (dashed line) clearly represents the measured data points. In the low-O:C case, LLEVAP represents the early stages of evaporation better while the later part is closer to the KM-GAP curve. In summary, the assumption of liquid like behaviour for RH40% is valid, i.e., the viscosity at RH40% is still low enough for particle phase diffusion not to significantly limit the evaporation.

**1.41.5 SI References**

[revised manuscript text omitted]

**Figure S5: Individual O:C ratios of the detected molecules in normalised integrated mass spectra for dry, fresh SOA particles in low-, medium- and high-O̅:C̅ cases. All ions with the same O:C ratio for a given carbon chain length were added up. Symbol size indicates signal strength for the dry, fresh SOA case, and colour code illustrates the changes due to isothermal evaporation under dry conditions (panels (a), (c), and (e)) and between fresh SOA under dry and RH80% conditions (panels (b), (d), and (f)). Red colours indicate higher contributions in the fresh, dry case while blue colours indicate a net increase with evaporation or humidification.**

[Figure]

**Figure S6: Non-normalised integrated mass spectra of all filter blank measurements for low-σ:C (panels (a) and (b)) and non-normalised total thermograms (c) for filter blanks and measurements after evaporation in the RTC. The colour code is the same in all three panels.**

[Figure]

**Figure S7:** Non-normalised integrated mass spectra for highest (a) and lowest (b) mass loading on the FIGAERO filter in the high-O:C cases. Panel (c) shows the first filter blank measurement in the morning of that experiment day (i.e. maximum background).